# Prevalence, risk factors and consequences of newborns born small for gestational age: a multisite study in Nepal

Pragya Gautam Paudel,[1,2] Avinash K Sunny,[2] Rejina Gurung,[2] Abhishek Gurung,[2] Honey Malla,[2] Shyam Sundar Budhathoki,[2,3] Prajwal Paudel,[4] Navraj KC,[4] Ashish KC  [5]

► Additional material is published online only. To view please visit the journal online (http://dx.doi.org/10.1136/bmjpo-2019-000607).

[1]Department of Public Health, University of Tennessee Knoxville, Knoxville, Tennessee, USA
[2]Research Division, Golden Community, Lalitpur, Nepal
[3]Department of Public Health, Imperial College London, London, UK
[4]Department of Public Health, Government of Nepal Ministry of Health and Population, Kathmandu, Nepal
[5]Department of Women's and Children's Health, Uppsala University, Uppsala, Sweden

**Correspondence to**
Dr Ashish KC; aaashis7@yahoo.com

## ABSTRACT

**Objective** To identify the prevalence, risk factors and health impacts associated with small for gestational age (SGA) births in Nepal.

**Methods** A cross-sectional study was conducted in 12 public hospitals in Nepal from 1 July 2017 to 29 August 2018. A total of 60 695 babies delivered in these hospitals during the study period were eligible for inclusion. Clinical information of mothers and newborns was collected by data collectors using a data retrieval form. A semistructured interview was conducted at the time of discharge to gather sociodemographic information from women who provided the consent (n=50 392). Babies weighing less than the 10th percentile for their gestational age were classified as SGA. Demographic, obstetric and neonatal characteristics of study participants were analysed for associations with SGA. The association between SGA and likelihood of babies requiring resuscitation or resulting in stillbirth and neonatal death was also explored.

**Results** The prevalence of SGA births across the 12 hospitals observed in Nepal was 11.9%. After multiple variable adjustment, several factors were found to be associated with SGA births, including whether mothers were illiterate compared with those completing secondary and higher education (adjusted OR (AOR)=1.73; 95% CI 1.09 to 2.76), use of polluted fuel compared with use of clean fuel for cooking (AOR=1.51; 95% CI 1.16 to 1.97), first antenatal care (ANC) visit occurring during the third trimester compared with first trimester (AOR=1.82; 95% CI 1.27 to 2.61) and multiple deliveries compared with single delivery (AOR=3.07; 95% CI 1.46 to 6.46). SGA was significantly associated with stillbirth (AOR=7.30; 95% CI 6.26 to 8.52) and neonatal mortality (AOR=5.34; 95% CI 4.65 to 6.12).

**Conclusions** Low literacy status of mothers, use of polluted fuel for cooking, time of first ANC visit and multiple deliveries are associated with SGA births. Interventions encouraging pregnant women to attend ANC visits early can reduce the burden of SGA births.

### What is known about the subject?

► In 2012, twenty-three million babies were born small for gestational age (SGA) in low and middle-income countries (LMIC).
► Globally, the burden of babies born SGA is high in LMICs and is substantially concentrated in South Asia.
► More than 20% of neonatal deaths in LMICs are attributable to SGA births.What this study adds?

### What this study adds?

► We found the prevalence of SGA births to be 11.9%.
► Illiteracy in mother, use of polluted fuel for cooking, time of first antenatal care visit during third trimester and multiple deliveries increased risk of SGA births

Strategy 2016–2030 aims to reduce the national neonatal mortality rate to 12 per 1000 births.[1] The Sustainable Development Goal 3 target for child mortality aims to end preventable deaths of newborns and children under 5 years of age by 2030.[2] The strategy strives for a world in which newborns have an equal opportunity to survive and thrive through childhood. However, in order to achieve this goal, we require the data to inform evidence on addressing the risk factors for small for gestational age (SGA) births. In 2012, twenty-three million babies were born SGA in low and middle-income countries (LMIC).[3] A study conducted by Lee *et al* in 2017 mentioned that the largest burden of SGA births was recorded in South Asia where prevalence and neonatal mortality related to SGA were 34% and 26%, respectively.[3] In Nepal, the estimated prevalence was 30%–40% in 2010.[4] The global burden of SGA accounts for one-third of low birth

## INTRODUCTION

The United Nations Global Strategy for Women's, Children's and Adolescent Health

weight.[4–6] Another study by Ota *et al* in 2014 found that the prevalence of SGA was 17.9% in Nepal.[7]

SGA births are babies born smaller in size than usual for their gestational age[8] and commonly defined as weighing less than the 10th percentile for the gestational age.[9–11] The importance of birth weight is exemplified by the Barker hypothesis which states that conditions during pregnancy will determine the long-term effects in later life.[12] The complications of SGA birth are not confined only to perinatal and neonatal outcomes as they are also linked to long-term adversities, including developmental complications.[13] Babies born SGA are more likely to have neonatal infections, perinatal respiratory complications, jaundice, polycythaemia, hypoglycaemia, poor feeding and hypothermia leading to increased likelihood of childhood deaths.[3 14] A study conducted by the Child Health Epidemiology Reference Group revealed significantly increased risk of neonatal and postneonatal mortality among babies born SGA.[3] More than 20% of neonatal deaths in LMICs were attributable to SGA births in 2012.[3]

Currently, there are not enough evidence specific to SGA births in Nepal. As SGA has considerable health impacts on the well-being of babies[15 16] identifying risk factors and consequences associated with SGA births in Nepal is imperative in order to build evidence to inform sustainable interventions which aim to mitigate the risk of adverse perinatal outcomes and associated long-term consequences of childhood development. This study aims to address this need through identifying the prevalence, risk factors and short-term health impacts associated with SGA babies across 12 hospitals in Nepal.

## METHODS
This study is reported in compliance with the Strengthening the Reporting of Observational Studies in Epidemiology checklist.

### Study design and setting
This is a cross-sectional study across 12 public hospitals in Nepal, conducted from 1 July 2017 to 29 August 2018. This study is nested within an evaluation of the scaling up of neonatal quality improvement project in Nepal.[17] The selected hospitals are from various geographical locations and cover both urban and rural areas in Nepal (table 1).

### Patient and public involvement
This research question aims to assess the burden of SGA birth and its risk factor. The improved care and monitoring for high-risk mother will reduce SGA birth. The results of the study will be disseminated to the study participants through local and national news media.

### Study population
All births with gestational age of 22 weeks or more delivered at these hospitals during the study period were included. Babies who were born outside these hospitals,

**Table 1** Background of the selected hospitals (year: 2015)*

| Name of hospital | Total deliveries per year | Services available |
|---|---|---|
| Western Regional Hospital | 9427 | L&D+OT+SNCU |
| Mid-Western Regional Hospital | 3139 | L&D+OT+SNCU |
| Bardiya District Hospital | 1065 | L&D+SNCU |
| Bharatpur Hospital | 11 318 | L&D+OT+NICU |
| Seti Zonal Hospital | 5767 | L&D+OT+SNCU |
| Nuwakot District Hospital | 1438 | L&D+OT+SNCU |
| Koshi Zonal Hospital | 8355 | L&D+OT+SNCU |
| Rapti Sub-Regional Hospital | 3280 | L&D+SNCU |
| Nawalparasi District Hospital | 1374 | L&D+SNCU |
| Lumbini Zonal Hospital | 9007 | L&D+OT+NICU |
| Bheri Zonal Hospital | 4276 | L&D+OT+SNCU |
| Pyuthan District Hospital | 1194 | L&D+OT+SNCU |

*Estimated total deliveries per year in each hospital are according to the year 2015, before the study protocol was published.[22]
L&D, labour and delivery room; NICU, neonatal intensive care unit; OT, operation theatre; SNCU, sick newborn care unit.

had major malformations or whose mothers did not provide consent were excluded from the study.

### Data collection and management
A surveillance system for systematic collection of data was established to collect maternal and newborn health data. The study had two distinct parts: (1) a chart review for which the ethical committee provided a waiver for consent, and (2) a semistructured interview for which individual consent was obtained (online supplementary file). All eligible women were admitted in the hospital for delivery and consented to be enrolled in the study; clinical information during labour, birth and postpartum period was collected from patient case note by data collectors using a data retrieval form. A waiver of informed consent for health record data was obtained prior to data collection. A semistructured interview was conducted at the time of discharge to gather sociodemographic and antenatal care (ANC) information from women who provided the consent. Following data extraction and interviews, completed forms were assessed by the data coordinator. To ensure accuracy of data collection, 10% of mothers' information was recollected in a random fashion to ensure the consistency and completeness of data collection. These information sheets were indexed at the end of each day by the data coordinator. Every week, the completed forms were sealed in an opaque envelope and sent to the Kathmandu office for further data management (storing, sorting, entry and cleaning). The data were also assessed for completeness.

Data cleaning was performed in Census and Survey Processing System every month. The cleaned data were exported into Statistical Package for the Social Sciences (SPSS) for data analysis. Personal patient information

was removed prior any data analysis. All hard copies of information sheet were indexed and stored in compliance with the ethical guidelines.

## Variables included in the study

The outcome variable included in this study was SGA which was calculated through measuring gestational age by last menstrual period (LMP) and birth weight (grams). We used the standard criteria of 10th percentile according to WHO and defined SGA in our study as a neonate with birth weight lower than the 10th percentile for babies of same gestational age, using appropriate for gestational age as a reference population.[7 18] The details regarding sociodemographic variables, obstetric variables and neonatal variables included in this study are explained in the online supplementary table 1.

## Statistical analysis

Prevalence of SGA was calculated as number of SGA cases per total number of births included in the study. Bivariate analysis was performed using binary logistic regression for sociodemographic, obstetric and neonatal risk factors. The clinically and contextually potential risk factors with p value <0.2 in the bivariate analysis were used to perform a multivariable logistic regression. Multicollinearity between the variables was checked before selecting the variables for the model. In addition to the risk factors, health impacts of SGA births including stillbirth, resuscitation and neonatal mortality were explored. During data analysis, multiple imputation was performed for missing values of education, maternal smoking, second-hand smoking and indoor smoking using sex of baby as variable weightage. Statistical analyses were performed using SPSS V.25.

## RESULTS

A total of 63 099 pregnant women were admitted to the 12 hospitals participating in the study. Out of these, 60 742 delivered during the study period. Out of these, 60 695 babies were eligible for the first part of the study, and 7221 births were classified as SGA (figures 1 and 2). The prevalence of SGA births within the hospitals was 11.9%.

## Sociodemographic, obstetric and neonatal factors

Table 2 describes bivariate analysis of sociodemographic, obstetric and neonatal factors associated with SGA births based on the chart review. Sociodemographic factors such as maternal ages of 15–19 years compared with mothers aged 20–34 years (crude OR (cOR)=1.40; 95% CI 1.31 to 1.49), obstetric factors such as nulliparous mothers compared with multiparous (cOR=1.13; 95% CI 1.06 to 1.21), multiple deliveries compared with single delivery (cOR=5.38; 95% CI 4.38 to 6.61), babies born to severe anaemic mothers (cOR=2.36; 95% CI 1.69 to 3.29) and female babies compared with male babies (cOR=1.41; 95% CI 1.34 to 1.48) are associated with SGA births (table 2). Significantly associated risk factors in

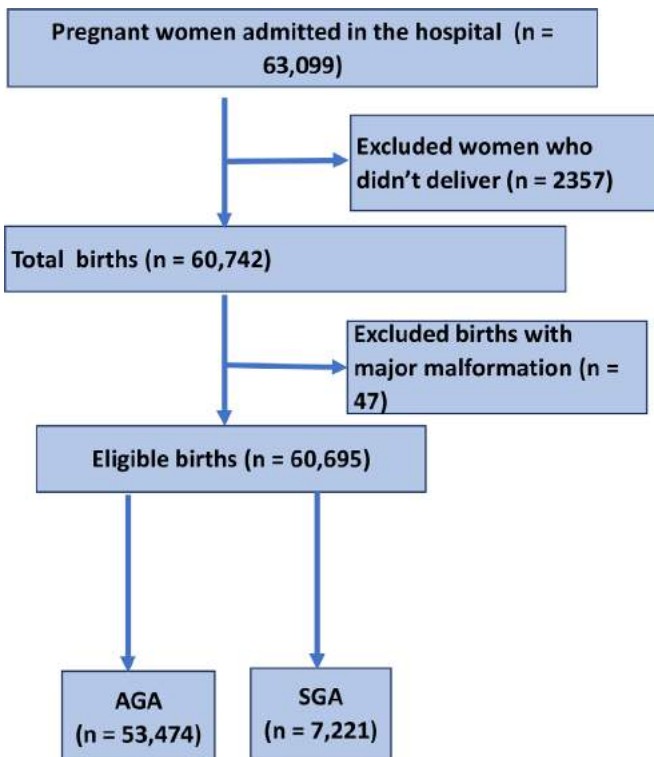

**Figure 1** Strengthening the Reporting of Observational Studies in Epidemiology (STROBE) flow diagram. AGA, appropriate weight for gestational age; SGA, small for gestational age.

bivariate analysis in table 2 were included in the multivariable analysis.

Table 3 describes the bivariate analysis of sociodemographic and obstetric factors associated with SGA birth based on interview. Mothers who have history of smoking

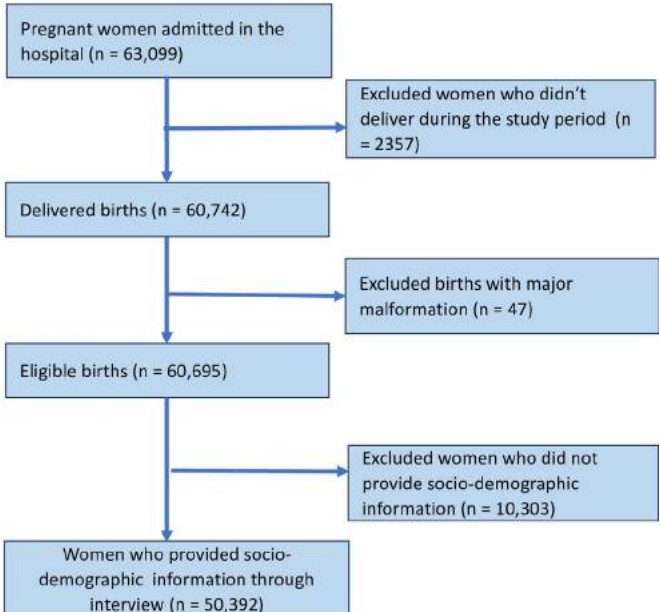

**Figure 2** Strengthening the Reporting of Observational Studies in Epidemiology (STROBE) flow diagram showing number of women who provided consent.

**Table 2** Sociodemographic, obstetric and newborn characteristics associated with SGA births (chart review)

| Characteristics | SGA (n=7221) | AGA (n=53474) | Total (n=60695) | cOR (95% CI) | P value |
|---|---|---|---|---|---|
| **Mothers' age (years) (n=60695)** | | | | | |
| Mean age±SD | 23.43±4.44 | 23.98±4.35 | 23.92±4.37 | | |
| 15–19 | 1276 (17.7%) | 7148 (13.4%) | 8424 (13.9%) | 1.40 (1.31 to 1.49) | <0.001 |
| 20–34 | 5751 (79.6%) | 44934 (84.0%) | 50685 (83.5%) | Reference | |
| 35 and above | 194 (2.7%) | 1392 (2.6%) | 1586 (2.6%) | 1.09 (0.94 to 1.27) | 0.274 |
| **Ethnicity (n=60695)** | | | | | |
| Dalit | 1398 (19.4%) | 9271 (17.3%) | 10669 (17.6%) | 0.95 (0.81 to 1.10) | 0.488 |
| Janajati | 1945 (26.9%) | 15675 (29.3%) | 17620 (29.0%) | 0.78 (0.67 to 0.91) | <0.001 |
| Madhesi | 672 (9.3%) | 3893 (7.3%) | 4565 (7.5%) | 1.08 (0.92 to 1.28) | 0.334 |
| Muslim | 220 (3.0%) | 1382 (2.6%) | 1602 (2.6%) | Reference | |
| Chhetri/Brahmin | 2717 (37.6%) | 21105 (39.5%) | 23822 (39.2%) | 0.81 (0.70 to 0.94) | 0.005 |
| Other castes | 269 (3.7%) | 2148 (4.0%) | 2417 (4.0%) | 0.79 (0.65 to 0.95) | 0.014 |
| **Parity (n=60695)** | | | | | |
| Multiparous (>2 births) | 1218 (16.9%) | 9183 (17.2%) | 10401 (17.1%) | Reference | |
| Nulliparous | 3883 (53.8%) | 25906 (48.4%) | 29789 (49.1%) | 1.13 (1.06 to 1.21) | <0.001 |
| Primiparous (1 birth) | 2120 (29.4%) | 18385 (34.4%) | 20505 (33.8%) | 0.87 (0.81 to 0.94) | <0.001 |
| **Deliveries (n=60695)** | | | | | |
| Single | 7064 (97.8%) | 53254 (99.6%) | 60318 (99.4%) | Reference | |
| Multiple | 157 (2.2%) | 220 (0.4%) | 377 (0.6%) | 5.38 (4.38 to 6.61) | <0.001 |
| **Anaemia (n=6057)** | | | | | |
| No | 696 (93.5%) | 5162 (97.2%) | 5858 (96.7%) | Reference | |
| Yes | 48 (6.5%) | 151 (2.8%) | 199 (3.3%) | 2.36 (1.69 to 3.29) | <0.001 |
| **Antepartum haemorrhage (n=60695)** | | | | | |
| No | 7170 (99.3%) | 53284 (99.6%) | 60454 (99.6%) | Reference | |
| Yes | 51 (0.7%) | 190 (0.4%) | 241 (0.4%) | 2.01 (1.46 to 2.72) | <0.001 |
| **Sex of baby** | | | | | |
| Male | 3353 (46.4%) | 29374 (54.9%) | 32727 (53.9%) | Reference | |
| Female | 3868 (53.6%) | 24100 (45.1%) | 27968 (46.1%) | 1.41 (1.34 to 1.48) | <0.001 |

AGA, appropriate weight for gestational age; cOR, crude OR; SGA, small for gestational age.

(cOR=1.10; 95% CI 1.01 to 1.20), mothers living with someone who smokes at home (cOR=1.17; 95% CI 1.10 to 1.25), mothers who used polluted fuel compared with clean fuel use (cOR=1.35; 95% CI 1.26 to 1.45) and mothers who are illiterate compared with secondary or higher education (cOR=1.55; 95% CI 1.38 to 1.74) were significantly associated with SGA births. Mothers who made less than 4 ANC compared with >4 ANC visits (cOR=1.30; 95% CI 1.22 to 1.39), mothers whose first ANC visit was during second (cOR=1.13; 95% CI 1.06 to 1.20) and third trimesters (cOR=1.40; 95% CI 1.29 to 1.51) compared with first trimester (first ANC visit) and mothers who had no prior delivery preparation (cOR=1.32; 95% CI 1.22 to 1.43) were significantly associated with SGA births. Further, significantly associated risk factors in bivariate analysis in table 3 were included in the multivariable analysis. The variables selected for the model did not show a multicollinearity problem.

## Multivariable analysis

Table 4 summarises the result of multivariable analysis. Mothers who were illiterate had 1.73 times higher odds of having SGA births compared with mothers who had secondary and above education (adjusted OR (AOR)=1.73; 95% CI 1.09 to 2.76). In comparison with use of clean fuel for cooking purpose, use of polluted fuel was associated with a higher risk of having SGA births (AOR=1.51; 95% CI 1.16 to 1.97). Mothers whose first ANC visit was during third trimester had 1.82 times higher odds of having SGA births compared with first ANC visit during the first trimester (AOR=1.82; 95% CI 1.27 to 2.61). Multiple deliveries were significantly associated with SGA births compared with single delivery (AOR=3.07; 95% CI 1.46 to 6.46). Girls had 1.29 times higher odds of being born SGA (AOR=1.29; 95% CI 1.08 to 1.56) as compared with boys.

**Table 3** Sociodemographic and obstetric characteristics associated with SGA births (interview)

| Characteristics | SGA (n=7221) | AGA (n=53474) | Total (n=60695) | cOR (95% CI) | P value |
|---|---|---|---|---|---|
| **Education (n=60525)** | | | | | |
| Illiterate | 440 (6.1%) | 2474 (4.6%) | 2914 (4.8%) | 1.55 (1.38 to 1.74) | <0.001 |
| Literate | 818 (11.4%) | 5370 (10.1%) | 6188 (10.2%) | 1.23 (1.13 to 1.25) | <0.001 |
| Basic education | 1166 (16.2%) | 8513 (16.0%) | 9679 (16.0%) | 1.14 (1.05 to 1.23) | <0.001 |
| Secondary education and above | 3476 (63.9%) | 30835 (68.6%) | 34311 (68.1%) | Reference | |
| **Smoking (n=60525)** | | | | | |
| No | 6349 (88.5%) | 47591 (89.2%) | 53940 (89.1%) | Reference | |
| Yes | 828 (11.5%) | 5757 (10.8%) | 6585 (10.9%) | 1.10 (1.01 o 1.20) | 0.032 |
| **Anyone in the same house smokes (n=60525)** | | | | | |
| No | 5215 (72.7%) | 39897 (74.8%) | 45112 (74.5%) | Reference | |
| Yes | 1962 (27.3%) | 13451 (25.2%) | 15413 (25.5%) | 1.17 (1.10 to 1.25) | <0.001 |
| **Type of fuel use for cooking (n=60525)** | | | | | |
| Clean | 2721 (37.9%) | 21639 (40.6%) | 24360 (40.3%) | Reference | |
| Polluted | 2033 (28.3%) | 12880 (24.1%) | 14913 (24.6%) | 1.35 (1.26 to 1.45) | <0.001 |
| Both clean and polluted fuel | 2423 (33.8%) | 18824 (35.3%) | 21247 (35.1%) | 1.02 (0.96 to 1.10) | 0.495 |
| **ANC visit (n=49867)** | | | | | |
| Four or more visits | 3901 (72.4%) | 34415 (77.4%) | 38316 (76.8%) | Reference | |
| Less than four visits | 1486 (27.6%) | 10065 (22.6%) | 11551 (23.2%) | 1.30 (1.22 to 1.39) | <0.001 |
| **Time of first ANC visit (n=49867)** | | | | | |
| First trimester | 2042 (37.9%) | 18806 (42.3%) | 20848 (41.8%) | Reference | |
| Second trimester | 2277 (42.3%) | 18627 (41.9%) | 20904 (41.9%) | 1.13 (1.06 to 1.20) | <0.001 |
| Third trimester | 1068 (19.8%) | 7047 (15.8%) | 8115 (16.3%) | 1.40 (1.29 to 1.51) | <0.001 |
| **Delivery preparation (n=50392)** | | | | | |
| Yes | 4587 (84.4%) | 39430 (87.7%) | 44017 (87.3%) | Reference | |
| No | 849 (15.6%) | 5526 (12.3%) | 6375 (12.7%) | 1.32 (1.22 to 1.43) | <0.001 |

AGA, appropriate weight for gestational age; ANC, antenatal care; cOR, crude OR; SGA, small for gestational age.

**Table 4**  Multivariable analysis of factors associated with SGA births

| Variables | βcoefficient | AOR (95% CI) | P value |
|---|---|---|---|
| **Maternal age** | | | |
| 20–34 years | Reference | | |
| 15–19 years | 0.16 | 1.18 (0.85 to 1.63) | 0.329 |
| 35 and above | −0.25 | 0.78 (0.48 to 1.28) | 0.325 |
| **Education** | | | |
| Secondary and above | Reference | | |
| Illiterate | 0.55 | 1.73 (1.09 to 2.76) | 0.020 |
| Literate | 0.11 | 1.11 (0.74 to 1.67) | 0.611 |
| Basic education | −0.02 | 0.98 (0.75 to 1.30) | 0.906 |
| **Smoking** | | | |
| No | Reference | | |
| Yes | −0.11 | 0.90 (0.62 to 1.32) | 0.586 |
| **Anyone in the same house smokes** | | | |
| No | Reference | | |
| Yes | 0.16 | 1.17 (0.89 to 1.52) | 0.246 |
| **Type of fuel used** | | | |
| Clean | Reference | | |
| Polluted | 0.41 | 1.51 (1.16 to 1.97) | 0.002 |
| Both clean and polluted | 0.31 | 1.36 (1.05 to 1.77) | 0.021 |
| **Parity** | | | |
| Multipara (>2 births) | Reference | | |
| Nullipara (never carried a pregnancy) | 0.14 | 1.15 (0.89 to 1.50) | 0.292 |
| Primipara (1 birth) | −0.22 | 0.80 (0.62 to 1.04) | 0.092 |
| **Anaemia** | | | |
| No | Reference | | |
| Yes | 0.25 | 1.29 (0.78 to 2.11) | 0.321 |
| **Deliveries** | | | |
| Single | Reference | | |
| Multiple | 1.12 | 3.07 (1.46 to 6.46) | 0.003 |
| **Antepartum haemorrhage** | | | |
| No | Reference | | |
| Yes | −0.45 | 0.64 (0.15 to 2.75) | 0.544 |
| **ANC visit during pregnancy** | | | |
| ≥4 | Reference | | |
| <4 | 0.16 | 1.18 (0.94 to 1.49) | 0.164 |
| **Time for first ANC visit** | | | |
| First trimester | Reference | | |
| Second trimester | 0.24 | 1.27 (0.99 to 1.63) | 0.063 |
| Third trimester | 0.60 | 1.82 (1.27 to 2.61) | 0.001 |
| **Delivery preparation** | | | |
| No | Reference | | |
| Yes | 0.15 | 1.16 (0.81 to 1.68) | 0.421 |
| **Sex of baby** | | | |
| Male | Reference | | |
| Female | 0.26 | 1.29 (1.08 to 1.56) | 0.006 |

These variables were adjusted with maternal age, education, ethnicity, smoking, anyone in the same house smokes, type of fuel use for cooking, parity, deliveries, anaemia, antepartum haemorrhage, antenatal care (ANC) visit, time of first ANC visit, delivery preparation and sex of baby.
AOR, adjusted OR; SGA, small for gestational age.

**Consequences of SGA births**

Tables 5 and 6 summarise the consequence in babies with SGA births. Babies born with SGA births were significantly associated with stillbirth (AOR=7.30; 95% CI 6.26 to 8.52) and neonatal mortality (AOR=5.34; 95% CI 4.65 to 6.12) adjusted with maternal age, multiple deliveries and sex of the baby.

**DISCUSSION**

This study aimed to identify the prevalence, risk factors and health impacts associated with SGA births in Nepal. In a large sample of hospital deliveries in Nepal, we found the prevalence of SGA births to be 11.9%. This finding is lower when compared with a study conducted in 2010, which revealed 30%–40% prevalence of SGA births in Nepal.[4] This study included a sample from WHO global survey on maternal and perinatal health between 2004 and 2008. Lower prevalence of SGA births in Nepal compared with previous finding could be due to an increase in women attending the hospitals for routine check-ups and improvement in quality of care in the hospitals in recent years. According to the Nepal Demographic and Health Survey (NDHS) 2016, 84% of pregnant women had at least one ANC contact with a skilled healthcare provider. The percentage of women who made four or more ANC visits increased from 50% in 2011 to 69% in 2016. Additionally, women's ANC visit during the first trimester increased by 15% in 2016 compared with 2011.[19] Another study by Ota *et al* in 2014 performed secondary analysis of the WHO multicountry survey and found that the prevalence of SGA was 17.9% in Nepal.[7] This study included a total sample of 10 474 from Nepal, among which 1874 were SGA births. Our study includes a larger sample from various public hospitals of Nepal that could have resulted in lower prevalence of SGA births.

Furthermore, a study conducted in LMICs revealed the prevalence of SGA births in LMICs was 19.3% in 2012.[3] Another study conducted in Tanzania showed the prevalence of SGA births was 16.6%.[20] The lower prevalence of SGA births in our study could be due to better awareness in women and higher ANC contact in recent years in Nepal. However, our study showed higher prevalence of SGA births when compared with a multicountry study conducted in other LMICs which revealed the prevalence of SGA births in Afghanistan, Uganda and Thailand as 4.8%, 6.6% and 9.7%, respectively.[7] This suggests that birth outcomes such as SGA could still be improved in Nepal.

Several potential risk factors associated with SGA births were explored in this study. Risk factors, such as illiteracy in mothers, use of polluted fuel for cooking, time of first ANC visit during third trimester, multiple deliveries and female babies, significantly increased the risk of SGA births. Studies conducted in Tanzania[15] and India[21] revealed that women's education reduced the risk of SGA births. These studies suggest that women with secondary or higher education mitigate the risk of SGA births.

**Table 5** Consequences of SGA in newborns

| Delivery outcome | Stillbirth (n=680) | Live birth (n=60 015) | cOR (95% CI) | P value |
|---|---|---|---|---|
| SGA | 332 (48.8%) | 6889 (11.5%) | 7.36 (6.32 to 8.57) | <0.001 |
| AGA | 348 (51.2%) | 53 126 (88.5%) | Reference | |
| **Resuscitation** | **Yes (n=12 416)** | **No (n=48 279)** | | |
| SGA | 1537 (12.4%) | 5684 (11.8%) | 1.06 (0.99 to 1.12) | 0.063 |
| AGA | 10 879 (87.6%) | 42 595 (88.2%) | Reference | |
| **Neonatal outcome** | **Alive at discharge (n=59 102)** | **Predischarge mortality (n=913)** | | |
| SGA | 6521 (11.0%) | 368 (40.3%) | 5.45 (4.76 to 6.23) | <0.001 |
| AGA | 52 581 (89.0%) | 545 (59.7%) | Reference | |

AGA, appropriate weight for gestational age; cOR, crude OR; SGA, small for gestational age.

Based on these findings, interventions to increase access to education for women may support efforts to reduce the burden of SGA births in countries such as Nepal. This paper found that the use of polluted fuel for cooking was associated with an increased risk of having SGA births. This finding is consistent with a study conducted in China,[22] which showed that use of biomass and coal for cooking was associated with greater risk of SGA births compared with the use of gas stove. This suggests the use of clean fuel, such as gas, as a cooking fuel may support in reducing the risk of SGA births.

This paper found significantly higher odds of SGA births in mothers whose time of first ANC visit was during the third trimester of pregnancy. This result was similar to the result obtained in a study conducted in Tanzania which revealed an increased risk of SGA births in women who were exposed to ANC for the first time during the third trimester of pregnancy.[15] Furthermore, multiple deliveries were significantly associated with SGA births. This finding is consistent with findings from studies conducted in Brazil,[23] USA[24] and India.[25]

**Table 6** Consequences of SGA adjusted with maternal age and infant risk fact

| Consequences | AOR (95% CI) | P value |
|---|---|---|
| **Delivery outcome (stillbirth)** | | |
| SGA | 7.30 (6.26 to 8.52) | <0.001 |
| Sex of baby (female=1) | 0.99 (0.85 to 1.15) | 0.851 |
| Maternal age (15–19 years) | 0.90 (0.72 to 1.13) | 0.372 |
| Maternal age (35 and above) | 2.79 (2.05 to 3.79) | <0.001 |
| Multiple deliveries (yes=1) | 1.86 (1.10 to 3.12) | 0.020 |
| **Neonatal death** | | |
| SGA | 5.34 (4.65 to 6.12) | <0.001 |
| Sex of baby (female=1) | 0.85 (0.74 to 0.97) | 0.013 |
| Maternal age (15–19 years) | 1.18 (0.99 to 1.41) | 0.071 |
| Maternal age (35 and above) | 2.24 (1.67 to 3.02) | <0.001 |
| Multiple deliveries (yes=1) | 2.92 (1.94 to 4.40) | <0.001 |

Stillbirth and neonatal deaths were adjusted for sex of baby, maternal age and multiple deliveries.
AOR, adjusted OR; SGA, small for gestational age.

Women with multiple pregnancies require special attention as well as regular and timely ANC to reduce the risk.[25]

We also identified that female babies had a higher risk of SGA births compared with male babies which was similar to the finding from a study that included LMICs.[26] Studies conducted in Brazil[23] and Nepal[27 28] also showed an increased risk of SGA births in female babies. However, this finding contradicts with a study conducted in Tanzania, which revealed that male babies were at higher risk of SGA births.[15] This study showed that male babies were 1.09 times at higher risk for SGA births compared with female babies. This could be due to a higher sample of male babies compared with female babies in their study while our study includes more female babies compared with male babies. Studies have shown an association between sex of the baby and SGA births, usually more likely in female babies[23 27 28]; however, the biological mechanism underlying this is unclear.[29]

In addition to the risk factors, this study also explored the health consequences related to SGA births specifically looking at delivery outcome, neonatal resuscitation and neonatal deaths in SGA births. Stillbirth and neonatal deaths were significantly higher among SGA births when adjusted for sex of baby, maternal age and multiple deliveries. The risk of stillbirth was higher among SGA babies, which agrees with results from a systematic review and meta-analysis, conducted in 2018, which showed a significant increase in risk of stillbirths among SGA infants.[30] Furthermore, we identified higher risk of neonatal mortality among SGA births. This result concurs with the result from a review study that included LMICs which demonstrated that there is an increased risk of neonatal mortality among SGA births.[4 20] Also, a study conducted by Lavin *et al* in South Africa showed significantly increased risk of stillbirths and neonatal mortality in SGA babies.[31] These findings suggest SGA as an important predictor of stillbirth and neonatal mortality, thus it is important to monitor fetal growth regularly. Early detection and management of SGA is essential to increase the chance of survival.

## Strengths and limitations

One major strength of this study is the large sample (60 695 eligible births) observed within 12 public hospitals across a large geographical scope in Nepal. This suggests the findings are representative of large proportion of births across Nepal. Another strength of this study is inclusion of all inborn babies, including stillbirths, which helped us explore the association between SGA and stillbirths.

There were several limitations within the study. Participant's information on height and weight was not available in patient case records and it was not feasible to measure the anthropometry during childbirth, so we could not assess the association of maternal anthropometry with SGA. Participant response to interview questions which require recalling past exposures to data collectors may result in recall and reporting bias. Additionally, the use of LMP in estimating the gestational age has several limitations, including uncertainty of LMP dates due to recall bias. It was also out of the scope of this study to follow-up babies after discharge to observe further health consequences. Another limitation associated with this study could be use of samples only from 12 hospitals which might not represent the overall population of the country. According to the 2016 NDHS, 57% of deliveries occur in health facilities.[19] Although this study includes a large sample from hospitals located in various geographical locations in Nepal, our study could not include those delivered outside the health facilities.

## CONCLUSION

Factors such as mothers' literacy status, use of polluted fuel for cooking, delay of first ANC visit, multiple deliveries and female babies are more likely to be associated with SGA births. Further, SGA increased the risk of stillbirth and neonatal mortality. Therefore, early identification of women with the above risk factors, implementation of programmes aiming to reduce the use of polluted fuel for cooking, prolonged education of female children and encouraging women to make early and regular ANC visits as some strategies could reduce the risks and impacts related to SGA births on babies' health and well-being.

**Acknowledgements** We thank Omkar Basnet, data manager, and all the data collectors. Also, we express our gratitude to the mothers who agreed to participate in the study.

**Contributors** PGP contributed on planning, creating new variables, performing data analysis on deidentified data and drafted the manuscript. AKS and AG performed data curation and data analysis and reviewed the first draft of the manuscript. RG, HM, SSB, PP and NKC reviewed the first draft of the manuscript and provided feedback. AKC conceptualised the study, provided the outline of the manuscript and reviewed the first draft and data analysis. All the authors read and approved the manuscript.

**Funding** This research was funded by the Swedish Research Council, Laerdal Foundation for Acute Medicine and Einhorn Family Foundation.

**Competing interests** None declared.

**Patient consent for publication** Obtained.

**Ethics approval** The study was approved by the ethical committee at Nepal Health Research Council (reference number 26-2017). A waiver for consent to use data from the hospital records was provided by the ethical committee. Informed consent was provided by 50 392 participants prior to obtaining sociodemographic information. A total of 10 303 women did not provide the consent for interview.

**Provenance and peer review** Not commissioned; externally peer reviewed.

**Data availability statement** Data are available upon reasonable request.

**ORCID iD**
Ashish KC http://orcid.org/0000-0002-0541-4486

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
