## [Reviewer comments · BMJ Paediatrics Open]

ARTICLE DETAILS

TITLE (PROVISIONAL)	Prevalence, risk factors and consequences of newborns born small for gestational age: a multi-site study in Nepal
AUTHORS	Gautam Paudel, Pragya; Sunny, Avinash; Gurung, Rejina; Gurung, Abhishek; Malla, Honey; Budhathoki, Shyam; Paudel, Prajwal; KC, Navraj; KC, Ashish

VERSION 1 – REVIEW

REVIEWER	Reviewer name: Peter Flom Institution and Country: Peter Flom Consulting USA Competing interests: None
REVIEW RETURNED	27-Nov-2019

GENERAL COMMENTS	I confine my remarks to statistical aspects of this paper. One issue is accounting for maternal size. I was puzzled that this is not included in the model and so, I did a little searching and found that, as I expected, maternal size is related to baby size but that, surprisingly (to me, anyway) the relationship was complicated. (I just sort of thought it would be natural that smaller women would have smaller babies). I think this needs some mention in the paper. Others may feel like I do. The definition of SGA as "bottom 10%tile" needs to be made more precise. Bottom 10%tile of what population? All babies in the world? Also, why use what could be called a norm based rather than a criterion based standard? That is, if we figure that maternal care and nutrition is generally improving over time, then the "bottom 10%" will get bigger and, presumably, less problematic. The percentile rank for a given weight and age will change, but the predicted health of the baby will (presumably) not. The prevalence of SGA was 11.9% (p. 2, line 29) which is pretty close to the 10%. This seems to mean that SGA is not a particular problem in Nepal. Moving to the analytic methods: The method of only including variables that were significant in bivariate analysis in the multivariate analysis is very common, but not correct. This is known as bivariate screening and it leads to many problems - all of the output from the final regression will be wrong. For details see, Regression Modeling Strategies by Frank Harrell. Briefly, the parameter estimates will be biased away from 0, the p values too small and the CIs too narrow. It is better to use substantive knowledge but if the authors insist on using an automatic method, LASSO is preferable. There appears to be quite a lot of missing data (judging based on
---

	the different Ns). Simply doing complete case analysis is only acceptable if the data were missing completely at random (MCAR) - that is, the reason for the data being missing had nothing to do with the dependent variable. If this is the case, then doing complete case analysis has lower power, but N here is large so that's not really an issue. But this needs to be stated and defended. If it is not MCAR then multiple imputation should probably be done.
--	---

REVIEWER	Reviewer name: Shalini Ojha Institution and Country: University of Nottingham, UK Competing interests: None
REVIEW RETURNED	None

GENERAL COMMENTS	Thank you for asking me to review this interesting manuscript. This is a multi-centre observational cohort study in Nepal investigating the associations of small gestational age (SGA) at birth with a number of maternal social and demographic factors and short term outcomes. The authors have explored an important research question. They found confirmation for some well-known associations of SGA. The study main strength is the large sample size spread over 12 centres across the country. I have some suggestions and questions: What the study adds section reads as a description of the study rather than statements of novel or interesting findings. Please summarise the results here with a focus on key findings in this section. Page 4 line 20: SGA, please give full forms of abbreviations when they first appear and use the abbreviations in subsequent uses. There is some inconsistency in this throughout the text. Page 4 line 35: Please replace normal for "usual" – SGA is a statistical definition and some SGA infants are normal. Page 5 line 5-10 sentence "Furthermore, stage of life" repeats information already given on Page 4 lines 49-53. Please delete one or the other. Page 5 lines 22 to 27: In the statement of the study aims, please specify "short term health impacts associated with SGA" The reference to the Barker's hypothesis followed by this statement leads one to expect results on short and long term outcomes, unless specified. Page 7 line 14-16: definition of SGA: please specify which neonatal or fetal growth charts were used to determine the 10th centile for gestational age. Page 7 line 27: Did the denominator include "all births" i.e. live and still born? Page 7 line 26: minor typo: heath in place of health Page 8 line 10: the reported prevalence of SGA is 11.9% - much lower than the 30-40% prevalence previously reported and quoted in the introduction. This is discussed in the first paragraph in the Discussion. I am not convinced that such a large reduction between 2010 and 2017-18 can have been solely due to increased antenatal contact with health care providers. Have the authors considered that there may be some selection bias in the study – they have recruited their cohort from hospitals. Are these institutions that can be accessed only by women of certain socio-economic status or other characteristics such that they have a lower risk giving birth to SGA infants? This is suggested by the 5% rate of illiteracy among the participants as compared to 33% among women aged 15-49 reported in the 2016 Nepal Demographic Health Survey.
--

	This lower prevalence of SGA needs further exploration and acknowledgment that it may not be representative of the whole population. Page 8 Socio-demographic, obstetric and neonatal factors and Table 2:  • Are these ORs or cORs? If these are cORs what factors have they been corrected for in these analysis as compared to the analyses in Table 3? • Line 25 – “smokes” instead of “smoke” • cOR quoted for maternal age is missing from the table • Suggestion re: “***” in Table 2 – these are redundant as the column heading says the figures are p values already. Same for tables 4 ad 5 Multivariate analysis (Page 9 and Table 3)  • Were all factors quoted in the section above included in the multivariate analysis? It is likely that several of these factors are strongly correlated with each other such as use of polluted fuel, Illiteracy, poor antenatal care. Did the authors account for multicollinearity before selecting factors for the model? • The final sentence in the paragraph may be more clear if written as “Girls had a 1.29 times higher odds of being born SGA as compared with boys”. Discussion Page 10 line 43-44: does more education reduce the risk of SGA birth or is it a combination of social and economic factors and health behaviours associated with women who are more educated that also reduce their chance of SGA. However, I do agree with this point, in general, and strongly endorse the next statement. My question is more about the scientific basis for this correlation and to discuss that it may not be one marker such as level of formal education but a more complex social and economic uplift that improves health outcomes such as SGA. Similar issues with use of polluted fuel. Additionally is there evidence that cooking stove smoke has an independent association with SGA (as there is for tobacco smoking). Page 11 line 40-42: please explain findings of ref 18 compared to this study further. It is unclear. The reasons for higher odds of SGA for one sex as compared to the other is interesting and requires further discussion
--	---

REVIEWER	Reviewer name: Peter Rohloffg Institution and Country: Maya Health Alliance - Guatemala Brigham and Women's Hospital - USA Competing interests: No competing interests
REVIEW RETURNED	12-Dec-2019

GENERAL COMMENTS	This is a remarkably ambitious study on an important topic, conducted within the context of a larger quality improvement initiative. I enjoyed reading it in detail. Overall the paper is well written, although it could still benefit from a close edit to correct minor errors of syntax and orthography. My primary questions have to do with clarifying the reference population used to define SGA and making sure that an appropriate reference population is defined and justified; and discussing in more detail the large amount of missing data. I have the following comments: Abstract: - I don't think the p values are necessary if CI are given
---

- In the Results, I would suggest mentioned something like “After multiple variable adjustment, several factors were found to be associated....”

- In the abstract conclusions the authors seem prepared to attributed causality to some factors which are almost certainly just markers of access and of poverty. For example, low education/literacy is a marker for risk and there is no assurance that ensuring girls stay in school will reduce SGA.

What this study adds:

- I would eliminate the “this study adds to the literature on risk factors” as not specific; Instead, could just expand on the third point, “In Nepal, factors associated with”

Introduction/Discussion: I do not know the literature on Nepal well. But I think there are a few other studies with data on SGA and associated factors (e.g., <https://www.ncbi.nlm.nih.gov/pubmed/25119107>, <https://www.ncbi.nlm.nih.gov/pubmed/25000447>). Might authors consider interacting with these studies here or in discussion so we have an even better sense of what this study adds?

Methods

- I feel the section on “data collection and management” could be edited for brevity and clarity. In particular, some of the details here seem appropriate for an ethics protocol but not a published paper (e.g. encryption of external hard drive).

- Missing is a definition of what reference population is used to define SGA. I think this is the most important omission in the paper. An article using data from Nepal has shown how much SGA analyses vary based on which reference population is used (<https://www.ncbi.nlm.nih.gov/pubmed/24642757>)

- Under ethics, given the diverse ethnicities and maternal languages involved, additional details on language of participants and researchers and how informed consent in this setting was adequately obtained would be helpful. This point comes up for me primarily because I note that in the Study flow diagram no individuals are noted as having declined to given informed consent (all births are included in the analysis).

Results

- Flow diagrams is clear, but again I’m surprised that there are no individuals who do not given signed informed consent. It would be helpful for the authors to confirm this point. Alternatively, if there are additional women who presented to hospital care in addition to the 63,099 women here who did not given informed consent, then those numbers should be included in the flow

- In description of the analyses and Table 2 and 4, cOR vs OR is used inconsistently. I suggest harmonizing this.

- With the presence of CI, I think p values could probably be eliminated in the text and just included in the table (Table 2, 3, 4) but this is a style thing.

- It looks like basically all variables in Table 2 are significant and therefore included in the multivariate model. Does this mean that other data on (nonsignificant) variables were collected and are not reported in this paper? If so, it would be helpful to modify the text and supplementary table to reference the complete list of variables - nonsignificance would be important, just as significance

- The exception to this is ethnicity which is significant in Table 2 (several have less risk) but this is not included in the multivariate

	analysis, please clarify!  - An important variable that is not included in the analysis is the hospital location of birth! I suspect that the different hospitals in this studies serve different ethnicities and have different case mix, and this should be analyzed. - There is a very substantial amount of missing data for most variables (>15%). I think some sensitivity analysis showing readers the distribution of missing data (e.g by hospital or ethnicity) and potentially multiple imputation would be helpful. - Discussion of Table 4 and 5 is fairly limited and should be expanded on. Also, it is unclear how the adjusted models in Table 5 were generated, i.e., why these and not different variables included here (c.f Table 2 to 3). Discussion/Conclusions  - Here I think the sentences comparing SGA in Nepal to high income countries are probably superfluous, since this is a given. It might be more useful to just interact with the details from LMICs, including add the few other studies with Nepal data as mentioned above - The discussion of factors associated with SGA needs a careful edit, because in a few places the authors seem inclined to attribute causality where this is not indicated. For example, low education is a marker for risk but it does not follow that increased education itself = less SGA
--	--

VERSION 1 – AUTHOR RESPONSE

Comment 1# I confine my remarks to statistical aspects of this paper. One issue is accounting for maternal size. I was puzzled that this is not included in the model and so, I did a little searching and found that, as I expected, maternal size is related to baby size but that, surprisingly (to me, anyway) the relationship was complicated. (I just sort of thought it would be natural that smaller women would have smaller babies). I think this needs some mention in the paper. Others may feel like I do.

Response-Thank you for the feedback. We could not determine the relationship of maternal size and size of the baby as there were a lot of missing data on height and weight of mothers during pregnancy. Therefore, it was difficult to assess the relationship of maternal size and size of baby. We have now mentioned this in the limitation as “the information of mother’s height and weight was not available in-patient case records and it was not feasible to measure the anthropometry during childbirth, so we could not assess the association of maternal anthropometry with SGA.”

Comment 2# The definition of SGA as "bottom 10%tile" needs to be made more precise. Bottom 10%tile of what population? All babies in the world? Also, why use what could be called a norm based rather than a criterion based standard? That is, if we figure that maternal care and nutrition is generally improving over time, then the "bottom 10%" will get bigger and, presumably, less problematic. The percentile rank for a given weight and age will change, but the predicted health of the baby will (presumably) not.

Response- Thank you for the comment. SGA was defined as neonates whose birth weight was lower than 10th percentile for babies of same gestational age. We have addressed this in page 7, line numbers 1 to 5. We used standard criteria of 10th percentile according to World Health Organization (WHO) which match our study population, meaning 10th percent of babies had low birth weight for the same gestational age.

Comment 3# The prevalence of SGA was 11.9% (p. 2, line 29) which is pretty close to the 10%. This seems to mean that SGA is not a particular problem in Nepal.

Response- Thank you for the comment. According to 2010, the estimated prevalence of SGA was 30% to 40%, which was based on the population-based survey. This is a hospital-based estimate using the intergrowth 21 as reference for the cut off of SGA. Another study that used multi-country study survey in 2014 showed the prevalence of 17.9%. This study included a total sample of 10,474 from Nepal. Among them small for gestational age (SGA) were 1,874. In our study, 7,221 out of 60,695 were SGA births. As our sample is larger which could be another reason for the lower percentage of SGA births in our study compared to previous study. As per your suggestion, we have addressed this in page 10, line 17 to 22 and page 11, line 1 to 16.

Comment 4# Moving to the analytic methods. The method of only including variables that were significant in bivariate analysis in the multivariate analysis is very common, but not correct. This is known as bivariate screening and it leads to many problems - all of the output from the final regression will be wrong. For details see, Regression Modeling Strategies by Frank Harrell. Briefly, the parameter estimates will be biased away from 0, the p values too small and the CIs too narrow. It is better to use substantive knowledge but if the authors insist on using an automatic method, LASSO is preferable.

Response- We agree with your comment. Now we have revised and included the variables with p-value <0.2 in multivariate analysis. Thus, we included not only the significant variables in bivariate analysis but also other potential variables with p-value <0.2. We have addressed this in page 7, line 9 and 10.

Comment 5# There appears to be quite a lot of missing data (judging based on the different Ns). Simply doing complete case analysis is only acceptable if the data were missing completely at random (MCAR) - that is, the reason for the data being missing had nothing to do with the dependent variable. If this is the case, then doing complete case analysis has lower power, but N here is large so that's not really an issue. But this needs to be stated and defended. If it is not MCAR then multiple imputation should probably be done.

We have now revised as per your suggestions performing multiple imputation for missing values. (Page 7, line 13 and 14).

Response to second reviewer's comment

Comment 6# What the study adds section reads as a description of the study rather than statements of novel or interesting findings. Please summarise the results here with a focus on key findings in this section.

Response- Thank you for the suggestion. We have revised this accordingly in page 3, line 9 to 13.

Comment 7# Page 4 line 20: SGA, please give full forms of abbreviations when they first appear and use the abbreviations in subsequent uses. There is some inconsistency in this throughout the text.

Response- Thank you so much for the suggestion. We have now revised this throughout the text.

Comment 8# Page 4 line 35: Please replace normal for "usual" – SGA is a statistical definition and some SGA infants are normal.

Response- Thank you for the suggestion. We have addressed this in page 4, line 15.

Comment 9# Page 5 line 5-10 sentence “Furthermore, stage of life” repeats information already given on Page 4 lines 49-53. Please delete one or the other.

Response- Thank you for the suggestion. We have addressed this.

Comment 10# Page 5 lines 22 to 27: In the statement of the study aims, please specify “short term health impacts associated with SGA” The reference to the Barker’s hypothesis followed by this statement leads one to expect results on short and long term outcomes, unless specified.

Response- Thank you for the suggestion. We have addressed this in line 8 to 10.

Comment 11# Page 7 line 14-16: definition of SGA: please specify which neonatal or fetal growth charts were used to determine the 10th centile for gestational age.

Response- We used standard definition of WHO to determine 10th percentile for gestational age (page 7, line 1 to 4).

Comment 12# Page 7 line 27: Did the denominator include “all births” i.e. live and still born?

Response- We included all births in the denominator.

Comment 13# Page 7 line 26: minor typo: heath in place of health

Response- Thank you. We have addressed this in page 7, line 12.

Comment 14# Page 8 line 10: the reported prevalence of SGA is 11.9% - much lower than the 30-40% prevalence previously reported and quoted in the introduction. This is discussed in the first paragraph in the Discussion. I am not convinced that such a large reduction between 2010 and 2017-18 can have been solely due to increased antenatal contact with health care providers. Have the authors considered that there may be some selection bias in the study – they have recruited their cohort from hospitals. Are these institutions that can be accessed only by women of certain socio-economic status or other characteristics such that they have a lower risk giving birth to SGA infants? This is suggested by the 5% rate of illiteracy among the participants as compared to 33% among women aged 15-49 reported in the 2016 Nepal Demographic Health Survey. This lower prevalence of SGA needs further exploration and acknowledgment that it may not be representative of the whole population.

Response- Your concern is genuine. We have taken all the births in these hospitals which suggests that there is less likely to be selection bias. However, 2016 Nepal Demographic and Health Survey (NDHS) is a survey conducted through multi stage random interviews. According to NDHS, we have only 57% of deliveries occur in health facilities. We have included all births occurred in 12 public hospitals located in various geographical locations in Nepal. Even though our study covers large population of the country, we could not cover the deliveries occurred outside the health facilities.

We have addressed this in page 10, line 9 to 14; page 10, line 1 to 16; page 13, line 1 to 6.

Comment 15# Page 8 Socio-demographic, obstetric and neonatal factors and Table 2: Are these ORs

or cORs? If these are cORs what factors have they been corrected for in these analysis as compared to the analyses in Table 3?

Line 25 – “smokes” instead of “smoke”

Response- These are cORs that have been adjusted in the multivariable analysis for all variables with $p < 0.2$ in bivariate analysis. These variables were adjusted with maternal age, education, ethnicity, smoking, anyone in the same house smoke, type of fuel use for cooking, parity, deliveries, anemia, antepartum hemorrhage, antenatal care (ANC) visit, time of first ANC visit, delivery preparation, sex of baby.

We have addressed the typo of “smokes” as “smoke” page 8, line 7.

Comment 16# cOR quoted for maternal age is missing from the table

Response- We have calculated cOR as categorical maternal age as included in table 2.

Comment 17# Suggestion re: “***” in Table 2 – these are redundant as the column heading says the figures are p values already. Same for tables 4 and 5
Multivariate analysis (Page 9 and Table 3)

Response- We have revised this as per your suggestion.

Comment 18# Were all factors quoted in the section above included in the multivariate analysis? It is likely that several of these factors are strongly correlated with each other such as use of polluted fuel, Illiteracy, poor antenatal care. Did the authors account for multicollinearity before selecting factors for the model?

Response- In multivariable analysis, we included all variables with $p < 0.2$ in bivariate analysis We explored the multicollinearity among the variables before selecting them for the model, as shown in page 7, line 10 and 11; page 8, line 20 and 21.

Comment 19# The final sentence in the paragraph may be more clear if written as “Girls had a 1.29 times higher odds of being born SGA as compared with boys”.

Thank you for the suggestion. We have addressed this in page 9, line 8 and 9.

Comment 20# Discussion. Page 10 line 43-44: does more education reduce the risk of SGA birth or is it a combination of social and economic factors and health behaviours associated with women who are more educated that also reduce their chance of SGA. However, I do agree with this point, in general, and strongly endorse the next statement. My question is more about the scientific basis for this correlation and to discuss that it may not be one marker such as level of formal education but a more complex social and economic uplift that improves health outcomes such as SGA. Similar issues with use of polluted fuel. Additionally is there evidence that cooking stove smoke has an independent association with SGA (as there is for tobacco smoking).

Response- We agree with your comment. In context of Nepal, there may not be only one marker such as level of education, but more complex social and economic uplift could improve the outcomes such as SGA. Economic factor can also play important role. However, we do not have complete data regarding the economic condition for all pregnant women. There are several missing cases, thus, we could not include in our study.

Usually, people with stronger economic condition and higher level of education use clean fuel, the complex uplift of these factors could improve health outcomes such as SGA.

Comment 21# Page 11 line 40-42: please explain findings of ref 18 compared to this study further. It is unclear. The reasons for higher odds of SGA for one sex as compared to the other is interesting and requires further discussion.

Response- Thank you for the suggestion. We have revised this as per your suggestion (page 11, line 14 to 23).

Response to third reviewer's comment

Comment 22# Abstract: I don't think the p values are necessary if CI are given

Response- Thank you for the suggestion. We have addressed this.

Comment 23# In the Results, I would suggest mentioned something like "After multiple variable adjustment, several factors were found to be associated...."

Response- Thank you for the suggestion. We have addressed this in page 2, line 12.

Comment 24# In the abstract conclusions the authors seem prepared to attributed causality to some factors which are almost certainly just markers of access and of poverty. For example, low education/literacy is a marker for risk and there is no assurance that ensuring girls stay in school will reduce SGA.

Response- We have revised this as per your suggestions in page 3, line 1 and 2.

Comment 25# What this study adds:- I would eliminate the "this study adds to the literature on risk factors" as not specific; Instead, could just expand on the third point, "In Nepal, factors associated with"

Response- Thank you so much for the feedback. We have addressed this as per your suggestion, page 3, line 9 to 13.

Comment 26# Introduction/Discussion: I do not know the literature on Nepal well. But I think there are a few other studies with data on SGA and associated factors (e.g., <https://www.ncbi.nlm.nih.gov/pubmed/25119107>, <https://www.ncbi.nlm.nih.gov/pubmed/25000447>). Might authors consider interacting with these studies here or in discussion so we have an even better sense of what this study adds?

Response- Thank you for suggestion. We have included the information in introduction section (page 4, line 13 and 14) and discussion section (page 9, line 19 to 22; page 10, line 12 to 15).

Comment 27# Methods-I feel the section on "data collection and management" could be edited for brevity and clarity. In particular, some of the details here seem appropriate for an ethics protocol but not a published paper (e.g, encryption of external hard drive).

Response- We have addressed this as per your suggestion.

Comment 28# Missing is a definition of what reference population is used to define SGA. I think this is the most important omission in the paper. An article using data from Nepal has shown how much SGA analyses vary based on which reference population is used (<https://www.ncbi.nlm.nih.gov/pubmed/24642757>)

Response- Thank you. We have addressed this in page 7, line 3 to 4.

Comment 29# Under ethics, given the diverse ethnicities and maternal languages involved, additional details on language of participants and researchers and how informed consent in this setting was adequately obtained would be helpful. This point comes up for me primarily because I note that in the Study flow diagram no individuals are noted as having declined to give informed consent (all births are included in the analysis).

Response- Thank you. The informed written consent form in Nepali language was used for the interviews with mothers before discharge. The study flow diagram includes all births in the analysis. Clinical information of mothers and newborns were collected using a data retrieval form from the maternity ward in the hospitals. A semi-structured interview was conducted at the time of discharge to gather socio-demographic and antenatal care information for which informed written consent written in Nepali language was used. We have addressed this in another flow diagram (Figure 2).

Comment 30# Results. Flow diagrams is clear, but again I'm surprised that there are no individuals who do not give signed informed consent. It would be helpful for the authors to confirm this point. Alternatively, if there are additional women who presented to hospital care in addition to the 63,099 women here who did not give informed consent, then those numbers should be included in the flow.

Response- Flow diagram (Figure 1) is used for the extraction of the clinical information of mothers and newborns from the data retrieval form from the maternity ward in the hospitals. We have added another flow diagram (Figure 2) to show the number of women who provided consent. A semi-structured interview was conducted at the time of discharge to gather socio-demographic and antenatal care information for which informed written consent written in Nepali language was used.

Comment 31# In description of the analyses and Table 2 and 4, cOR vs OR is used inconsistently. I suggest harmonizing this.

Response- Thank you for the suggestion. We have addressed this.

Comment 32# With the presence of CI, I think p values could probably be eliminated in the text and just included in the table (Table 2, 3, 4) but this is a style thing.

Response- Thank you for the suggestion. We have eliminated the p values from the text.

Comment 33# It looks like basically all variables in Table 2 are significant and therefore included in the multivariate model. Does this mean that other data on (nonsignificant) variables were collected and are not reported in this paper? If so, it would be helpful to modify the text and supplementary table to reference the complete list of variables - nonsignificance would be important, just as significance

Response- Thank you for the suggestion. We have revised and included the variables with p-value <0.2 in multivariate analysis. Thus, we included not only the significant variables but also other potential variables with p-value <0.2.

Comment 34# The exception to this is ethnicity which is significant in Table 2 (several have less risk) but this is not included in the multivariate analysis, please clarify!

Response- We included ethnicity; however, this was not significant.

Comment 35# An important variable that is not included in the analysis is the hospital location of birth! I suspect that the different hospitals in this studies serve different ethnicities and have different case mix, and this should be analyzed.

Response- In this study, we included the potential risk factors and we did not include by location of hospital since we covered 12 public hospitals in Nepal which provide similar care and serve similar ethnic groups.

Comment 36# There is a very substantial amount of missing data for most variables (>15%). I think some sensitivity analysis showing readers the distribution of missing data (e.g by hospital or ethnicity) and potentially multiple imputation would be helpful.

Response-We have now done the multiple imputations for missing value of education, smoking, anyone in the same house smoke and type of fuel use for cooking.

Maternal education			
Before imputation	AGA (N-47,192)	SGA (N-5900)	Total (N-53092)
Illiterate	2474 (4.6%)	440 (6.1%)	2914 (4.8%)
Literate	5370 (10.1%)	818 (11.4%)	6188 (10.2%)
Basic education	8513 (16.0%)	1166 (16.2%)	9679 (16.0%)
Secondary education and above	30835 (68.6%)	3476 (63.9%)	34311 (68.1%)
After imputation	AGA (N-53348)	SGA (N-7177)	Total-60525
Illiterate	2474 (4.6%)	440 (6.1%)	2914 (4.8%)
Literate	5370 (10.1%)	818 (11.4%)	6188 (10.2%)
Basic education	8513 (16.0%)	1166 (16.2%)	9679 (16.0%)
Secondary and above	36991 (69.3%)	4753 (66.2%)	41744 (69.0%)
Smoking			
before imputation	AGA (N-44955)	SGA (N-5435)	Total (N-50390)
No	39934 (88.8%)	4775 (87.9%)	44709 (88.7%)
Yes	5021 (11.2%)	660 (12.1%)	5681 (11.3%)
After imputation	AGA (N-53348)	SGA (N-7177)	Total-60525
Smoking	AGA (N-53348)	SGA (N-7177)	Total-60525
No	47591 (89.2%)	6349 (88.5%)	53940 (89.1%)
Yes	5757 (10.8%)	828 (11.5%)	6585 (10.9%)
Somebody_at_home			
Before imputation	AGA (N-44955)	SGA (N-5435)	Total (N-50390)
No	33545 (74.6%)	3888 (71.5%)	37433 (74.3%)
Yes	11411 (25.4%)	1548 (28.5%)	12959 (25.7%)

After imputation	AGA (N-53348)	SGA (N-7177)	Total-60525
No	39897 (74.8%)	5215 (72.7%)	45112 (74.5%)
Yes	13451 (25.2%)	1962 (27.3%)	15413 (25.5%)
Type of fuel use for cooking			
Before imputation	AGA (N-44955)	SGA (N-5435)	Total (N-50390)
Clean	21639 (40.6%)	2721 (37.9%)	24360 (40.3%)
Polluted	12880 (24.1%)	2033 (28.3%)	14913 (24.6%)
Both clean and polluted fuel	18824 (35.3%)	2423 (33.8%)	21247 (35.1%)
After imputation	AGA (N-53348)	SGA (N-7177)	Total-60525
Clean fuel used	21639 (40.6%)	2721 (37.9%)	24360 (40.3%)
Polluted fuel used	12880 (24.1%)	2033 (28.3%)	14913 (24.6%)
Both clean and polluted	18824 (35.3%)	2423 (33.8%)	21247 (35.1%)

Comment 37# Discussion of Table 4 and 5 is fairly limited and should be expanded on. Also, it is unclear how the adjusted models in Table 5 were generated, i.e., why these and not different variables included here (c.f Table 2 to 3).

Response- As per your suggestion, we have addressed this in page 12, line 1 to 13. We adjusted SGA with sex of the baby, maternal age, multiple deliveries for still birth and neonatal death before discharge, as shown in table 5.

Comment 38# Discussion/Conclusions-Here I think the sentences comparing SGA in Nepal to high income countries are probably superfluous, since this is a given. It might be more useful to just interact with the details from LMICs, including add the few other studies with Nepal data as mentioned above

Response- We have revised this as per your suggestion (page 10, line 9 to 16).

Comment 39# The discussion of factors associated with SGA needs a careful edit, because in a few places the authors seem inclined to attribute causality where this is not indicated. For example, low education is a marker for risk but it does not follow that increased education itself less SGA

Response- Thank you for the suggestion. While comparing with prior studies, we have mentioned as found in their articles; however, for interacting with other studies and concluding from we have mentioned as could or may. We have addressed this.

VERSION 2 – REVIEW

REVIEWER	Reviewer name: Peter Rohloff Institution and Country: Maya Health Alliance - Guatemala Brigham and Women's Hospital - USA Competing interests: No competing interests
REVIEW RETURNED	21-Jan-2020

GENERAL COMMENTS

The authors have done a lot of work and the paper is much improved.

I still have some additional comments however:

- In the abstract and conclusions, the authors still make causal statements. E.g., women with less education have more SGA, and therefore improving women's education = less SGA. I fear I have to insist on this point, because there are lots of (unmeasured) reasons that women with less education have higher risk for SGA. Authors should limit themselves to the observations made here about demographic and clinical characteristics associated with SGA.

- What this study adds: The first statement should be about the prevalence of sGA in this large hospital based sample (not Nepal as a whole)

- The study design is clearer now but I still have some questions. It is now clear to me that some data (clinical , chart review data) was collected on all patients in the hospitals. However, additional sociodemographic and exposure data was collected at discharge with informed consent. I think this means that the clinical data from the health records was collected without informed consent. This would be ok, if this issue was specifically addressed by the ethics committee - typically this would be using a waiver of informed consent mechanism for health records data. If this was in fact what happened, this needs to be clearer to the readers. On the contrary, if no waiver of informed consent was explicitly approved the usual procedure would be to analyze only the data for which informed consent was available.

- In one of the reviewer responses, the authors mention using the INTERGROWTH study as their reference population. However this is still not mentioned in the manuscript, and was my most significant concern on the first review. Readers need to know what reference population is being used, and why this reference population was chosen as opposed to another needs to be justified.

- Under statistical analysis, I note the statistical reviewers concerns about the model construction (bivariate screening). The authors have only partially addressed this by using a more generous p value cut off, but again the preferred method here is to construct a model based on clinical and contextual knowledge (rather than p value screening). It would be helpful for the authors to address this point specifically.

- Under the Table and flow chart review, it appears that 50K women gave informed consent. Even if the above is true and the authors got some data from chart review under a waiver of informed consent, it is unclear what then the informed consent was for. However, my reading is that a lot of data was collected on the discharge visit, meaning the denominator for many of the variables should be around 50K and not 60K (based on who gave informed consent). This issue definitely needs to be clarified.

- I'm still very interested in the ethnic and hospital case mix heterogeneity. From a study procedure perspective, how was consent handled for people not primarily speaking Nepali?

	From a discussion perspective, there is wide variability in SGA by ethnicity, please discuss. From an analysis perspective the hospitals are geographical diverse and have much different size/case load. I think hospital as a variable should be explored in the models.
--	--

VERSION 2 – AUTHOR RESPONSE

Response to first reviewer's comment

Comment 1# In the abstract and conclusions, the authors still make causal statements. E.g., women with less education have more SGA, and therefore improving women's education = less SGA. I fear I have to insist on this point, because there are lots of (unmeasured) reasons that women with less education have higher risk for SGA. Authors should limit themselves to the observations made here about demographic and clinical characteristics associated with SGA.

Response: We have revised this as per your suggestion in page 3 line 2.

Comment 2# What this study adds: The first statement should be about the prevalence of sGA in this large hospital-based sample (not Nepal as a whole)

Response: Thank you. We have addressed this as per your suggestion in page 3, line 13 and 14.

Comment 3# The study design is clearer now but I still have some questions. It is now clear to me that some data (clinical , chart review data) was collected on all patients in the hospitals. However, additional sociodemographic and exposure data was collected at discharge with informed consent. I think this means that the clinical data from the health records was collected without informed consent. This would be ok, if this issue was specifically addressed by the ethics committee - typically this would be using a waiver of informed consent mechanism for health records data. If this was in fact what happened, this needs to be clearer to the readers. On the contrary, if no waiver of informed consent was explicitly approved the usual procedure would be to analyze only the data for which informed consent was available.

Response: All eligible women who consented to the study at admission, clinical information is retrieved from the case study. We have mentioned this in page 7, line 8-10 as "All eligible women who were admitted in the hospital for delivery and consented to be enrolled in study, clinical information during labour, birth and postpartum period was collected from patient case note by data collectors using a data retrieval form"

Comment 4# In one of the reviewer responses, the authors mention using the INTERGROWTH study as their reference population. However, this is still not mentioned in the manuscript, and was my most significant concern on the first review. Readers need to know what reference population is being used, and why this reference population was chosen as opposed to another needs to be justified.

Response: We have now referenced intergrowth study, which has been used to make the cut off for SGA in the study in page 7, lines 9-10 as "We used the reference population of the intergrowth study to make the cut off for SGA and AGA[1]."

Comment 5# Under statistical analysis, I note the statistical reviewers concerns about the model construction (bivariate screening). The authors have only partially addressed this by using a more generous p value cut off, but again the preferred method here is to construct a model based on clinical and contextual knowledge (rather than p value screening). It would be helpful for the authors to address this point specifically.

Response: Thank you for your suggestion. We have revised this based on clinical and contextual knowledge in page 7 Line 11.

Comment 6# Under the Table and flow chart review, it appears that 50K women gave informed consent. Even if the above is true and the authors got some data from chart review under a waiver of informed consent, it is unclear what then the informed consent was for. However, my reading is that a lot of data was collected on the discharge visit, meaning the denominator for many of the variables should be around 50K and not 60K (based on who gave informed consent). This issue definitely needs to be clarified.

Response: We understand your concern. All women admitted in the hospital and who were enrolled in study are consented for the study. All 60K population consent was received for the study.

Comment 7# I'm still very interested in the ethnic and hospital case mix heterogeneity. From a study procedure perspective, how was consent handled for people not primarily speaking Nepali? From a discussion perspective, there is wide variability in SGA by ethnicity, please discuss. From an analysis perspective the hospitals are geographical diverse and have much different size/case load. I think hospital as a variable should be explored in the models.

Response: We understand your concern. Nepali is primary and official language spoken and written in all the hospitals of Nepal. When patients did not speak Nepali, their family members who speak Nepali informed them in local language.

We did not consider location of hospitals as a potential risk factor for SGA birth. Thus, we did not include in our analysis. Additionally, the included 12 public hospitals provide similar care. Also, all the ethnic groups in Nepal are proportionately scattered throughout the country. We explored ethnicity; however, we did not find association between ethnicity and SGA births.